# Regulation of AQP4 in the Central Nervous System

**DOI:** 10.3390/ijms21051603

**Published:** 2020-02-26

**Authors:** Arno Vandebroek, Masato Yasui

**Affiliations:** Department of Pharmacology, Keio University School of Medicine, 35 Shinanomachi, Shinjuku, Tokyo 160-8582, Japan; myasui@a3.keio.jp

**Keywords:** aquaporin-4, phosphorylation, microRNA, metal ion, water channel, small molecule inhibitor, TGN-020, acetazolamide, oocyte, proteoliposome

## Abstract

Aquaporin-4 (AQP4) is the main water channel protein expressed in the central nervous system (CNS). AQP4 is densely expressed in astrocyte end-feet, and is an important factor in CNS water and potassium homeostasis. Changes in AQP4 activity and expression have been implicated in several CNS disorders, including (but not limited to) epilepsy, edema, stroke, and glioblastoma. For this reason, many studies have been done to understand the various ways in which AQP4 is regulated endogenously, and could be regulated pharmaceutically. In particular, four regulatory methods have been thoroughly studied; regulation of gene expression via microRNAs, regulation of AQP4 channel gating/trafficking via phosphorylation, regulation of water permeability using heavy metal ions, and regulation of water permeability using small molecule inhibitors. A major challenge when studying AQP4 regulation is inter-method variability. A compound or phosphorylation which shows an inhibitory effect in vitro may show no effect in a different in vitro method, or even show an increase in AQP4 expression in vivo. Although a large amount of variability exists between in vitro methods, some microRNAs, heavy metal ions, and two small molecule inhibitors, acetazolamide and TGN-020, have shown promise in the field of AQP4 regulation.

## 1. Introduction

Aquaporins (AQPs) are a group of selective transmembrane channels which are expressed in every major organ group in the human body [1,2]. While 13 unique AQPs have been identified in the human body thus far, aquaporin-4 (AQP4) has become an interesting therapeutic target in various neurological disorders, due to its variety of functions and widespread expression in the nervous system [3]. AQP4 is the predominant water channel protein in the mammalian brain [4], and is expressed in astrocytes throughout the central nervous system [5]. In particular, AQP4 is located at the end-feet of astrocytic processes [6], which allows it to control nervous system water homeostasis via its involvement in water movement across the blood brain barrier [7]. AQP4 is a water specific channel, but has been shown to have various roles in neurotransmission [8,9,10], as well as possibly an effect on the activity of potassium channel Kir4.1 [11,12], although Kir4.1 has been shown to maintain functionality independent of AQP4 [13]. Loss of AQP4 has also been linked to altered levels of extracellular K^+^ [14], suggesting a role of AQP4 in K^+^ homeostasis.

AQP4 expression and activity in various neurological disorders has been examined, and several intriguing discoveries have been made. In cerebral edema and ischemia, AQP4 expression is increased [15,16], and inhibition of AQP4 activity, either by knockout or pretreatment with an inhibitor has been shown to cause a reduction in edema size and formation [7,17,18,19]. However, in vasogenic edema the loss of AQP4 activity actually aggravates the condition, due to the loss of ability to remove excess water from the brain tissue [2,5,16]. In glioblastomas and astrocytomas, AQP4 is upregulated and redistributed, showing a loss of polarized distribution in the end-feet of high grade tumors [12,20,21,22]. This loss of AQP4 polarity may contribute to increased migration capability of the astrocytoma cells, as AQP4 aggregation state has been shown to have an effect on glioma cell migration [23,24,25]. In addition, knockdown of AQP4 and specific expression of the orthogonal array of particles (OAP) forming isoform, M23, actually induced apoptosis in glioma cells [25,26], suggesting that the preferential expression of the AQP4 M1 isoform is important for the continued survival and spread of high grade brain tumors. While loss of AQP4 polarization during cell migration is a normal occurrence during reactive astrocyte migration [27], this usually occurs in response to brain injury, facilitating the formation of a glial scar [28]. AQP4 was found to be significantly upregulated in cases of temporal lobe epilepsy (TLE) via transcriptome analysis [29], and mislocalization of AQP4 is implicated as a contributing factor in this condition [30,31]. In addition to changes in expression or location of AQP4 in neurological disorders, AQP4 is also implicated as a target in neuromyelitis optica (NMO) [32,33,34], an autoimmune disease characterized by inflammation and demyelination of the optic nerve, large spinal cord lesions, and the presence of NMO-IgG in serum [35]. Finally, AQP4 has been shown to have many roles in Alzheimer’s disease, including possibly mediating the clearance of amyloid-β [36,37,38,39], and AQP4 deficiency leads to impaired learning and memory via disruption of both the glutamate transport pathway and K^+^ ion homeostasis [40,41,42,43]. With the increase in interest in AQP4 as a potential therapeutic target [1,5,16,19], and the implications AQP4 activity has on many neurological disorders [7,14,16,17,18,21,22,23,25,30,31,32,33,34,42,43], it is important to have a clear understanding of the myriad of ways in which AQP4 is known to be regulated.

## 2. Translational Regulation of AQP4

Regulation of AQP4 protein activity starts with regulation of gene expression. A developing field in AQP4 regulation is the application of microRNAs which target AQP4 [44]. MicroRNAs (miRNAs) are endogenous single-stranded majority intron-derived RNA sequences which specifically target messenger RNAs (mRNAs) for degradation or translation repression [45,46,47]. Several recent studies have shown an interaction between different miRNAs and AQP4, with therapeutic implications (Table 1). Knockdown of rat AQP4 (rAQP4) gene expression in vivo and mouse AQP4 (mAQP4) gene expression in vitro was shown to result in a significant increase in miR-224, which targets not only AQP4 but also gap junction protein connexin 43 (Cx43), which is one of the two primary proteins which form gap junction channels between astrocytes [48]. Knockdown of Cx43 resulted not only in a significant increase in miR-224 expression, but also in miR-19a, which also targets both proteins [48], implying that miR-19a and miR-224 may have important roles in the regulation of not only astrocyte connectivity but also water permeability. MiR-29b, which was already known to be downregulated in ischemic stroke, was shown to have a direct effect on expression of mAQP4 [49]. Upregulation of miR-29b correlated with a decrease in mAQP4 expression in ischemic mice, and reduced infarct volume, edema, and blood–brain barrier (BBB) disruption [49]. In normal conditions an upregulation of miR-29b can actually induce brain edema [50]. Exposure of Sprague-Dawley rats and CD-1 mice to 1,2-dichloroethane (1,2-DCE) resulted in an abnormal upregulation of miR-29b, a subsequent downregulation of rAQP4 and mAQP4, and the induction of brain edema [50], offering further evidence that miR-29b works by directly targeting AQP4 expression. Another miRNA associated with both AQP4 and ischemia is miR-145, which was found to attenuate rAQP4-induced astrocyte injury in an in vitro oxygen-glucose deprivation model of ischemia in primary cultured rat astrocytes [51]. A miRNA of particular interest which has been a point of interest in cerebral edema and glioma is miR-320a [52,53]. Mi-R320a expression was found to downregulate both rAQP4 and rAQP1 expression in conditions of cerebral edema [52], leading to an increase in infarct volume, while targeting miR-320a with anti-mi-R320a antibodies resulted in an increase in rAQP4 and rAQP1 expression with an accompanying decrease in infarct volume in vivo [52]. While downregulating the expression of miR-320a seems to be beneficial in cases of cerebral edema, in glioma cases this could have a detrimental effect, as miR-320a was found to inhibit glioma cell invasion and migration by targeting human AQP4 (hAQP4) [53]. Astrocytes rely on cell volume changes as an important tool in resizing the cell and allowing the cell to move through the intracellular space [25], inhibition of the activity of hAQP4 via miR-320a leads to a decrease in the ability of the glioma cells to invade and migrate in vitro, which has implications for cell migration in vivo [53]. In a similar fashion to miR-320a, targeting of miR-130a with anti-miR-130a antibodies has been shown to decrease edema size after cerebral ischemia in rats [54]. Interestingly, miR-130a has been shown to target the promotor of the M1 isoform of both rAQP4 and hAQP4, allowing for specific regulation of AQP4 M1, whereas miR-320a regulates all isoforms of AQP4 [54]. MiR-130b has been shown to have a protective effect in ischemic conditions, as upregulation of miR-130b protected against cerebral ischemic injury (CII) [55]. While many of these studies show a promising future for regulation of AQP4 by miRNAs, it is still unknown what complications artificial upregulation or downregulation of miRNAs may cause, and warrants further study.

## 3. Phosphorylation Driven Regulation of AQP4

Phosphorylation is a relatively well-studied post-translational protein modification, which can change the activity or expression of a protein by modulating protein structure, protein–protein interactions, and marking proteins for trafficking [56,57]. Phosphorylation of AQP4 has shown to play an important role in AQP4 trafficking and subcellular localization, as well as a potential role in channel gating [56,58,59,60]. Figure 1 shows an overview of identified AQP4 phosphorylation sites (in red) [56,59,61,62].

The subcellular localization of AQP4 can be controlled by phosphorylation at several serine residues in the protein [56,59]. Protein Kinase C (PKC) has been shown to downregulate hAQP4 by causing increasing levels of protein internalization, thereby reducing the amount of hAQP4 present on the cell membrane and the overall osmotic water permeability (Pf) of human gastric cells [56]. Significant downregulation of water permeability of AQP4 after treatment with PKC activators was found in several different models, including in vivo rat models [63], oocytes expressing rAQP4 [64], and human glioma cells [65]. This loss of water permeability was attributed to an internalization of rAQP4, as vasopressin induced rAQP4 endocytosis in oocytes [64]. The effect of PKC activators is likely due to PKC’s phosphorylation of the S180 residue of rAQP4, as a mutant form of rAQP4 (S180A) in which the serine was replaced with an alanine, showed no loss of water permeability [64]. However, in HEK293 cells and primary cultured astrocytes, AQP4 internalization was not shown to be dependent on PKC phosphorylation of S180, as mutant forms of hAQP4 (S180A and S180D) and treatment with PKC inhibitors did not show any difference in expression or translocation of hAQP4 [56,66]. As a possible alternative pathway for AQP4 internalization, phosphorylation by Protein Kinase A (PKA) was shown to induce hAQP4 and rAQP4 internalization in gastric HGT1 cells (transfected with rAQP4 cDNA) [58], HEK293 cells (transfected with hAQP4 cDNA), and rat primary cultured astrocytes [66]. This internalization is thought to be due to phosphorylation at S276, as a S276A mutant showed a loss of the internalization effect [66]. Interestingly, phosphorylation of S276, along with S285, T289, and S316 by casein kinase II (CKII) was shown to be necessary for Golgi transition in mouse primary cultured astrocytes, as mutations at these locations caused mAQP4 to localize to the Golgi bodies and lose the ability to migrate to the membrane in vitro [59]. These results indicate that if phosphorylation by CKII occurs at S276, mAQP4 is freed from the Golgi bodies and able to migrate to the cell membrane, but if the same phosphorylation site is activated by PKA, this directs the cell to internalize hAQP4 and rAQP4 and reduce their expression [58,59,66]. This contrast in results indicates that phosphorylation related AQP4 trafficking may be activated through many pathways which differ depending on the model used to study them.

A similarly complicated area of study is whether or not phosphorylation driven gating exists for AQP4 [56,60,61,67,68,69]. In rat astrocytes, water permeability through rAQP4 was found to be increased in the presence of extracellular lead [69], glutamate [67], and potassium [68]. This increase occurred without a corresponding change in the cellular location of rAQP4, implying that a gating type reaction is occurring [67,68,69]. In each case the water permeability change was attenuated by a protein kinase inhibitor; calcium dependent protein kinase II (CaMKII) inhibitors when cells were exposed to lead and glutamate, and PKA inhibitors for potassium exposure [67,68,69]. In the case of lead and glutamate, this reaction was identified to occur due to a phosphorylation of S111, as a S111A mutant did not show the same effect [67,69]. In addition, LLC-PK1 cells expressing green fluorescent protein (GFP) tagged mAQP4 which were treated with PKC activators and dopamine showed a loss of water permeability due to phosphorylation of S180 [70], which, in contrast with oocyte experiments suggesting that phosphorylation of S180 by PKC causes internalization of rAQP4 [64], did not show any noticeable protein distribution changes. Gating via phosphorylation is not unheard of in aquaporins, as both sheep AQP0 and hAQP1 have been suggested to have phosphorylation dependent gating pathways [71,72], and rAQP4 has already been shown to have a pH dependent gate at H95 (Figure 1, light blue residue), which increased water permeability in low pH in both simulated models and oocytes [73]. However, whether or not AQP4 contains a phosphorylation-dependent gate is still under debate. In oocyte based experiments, phosphorylation of rAQP4 S111 and mutation of S111 to S111A and S111D had no observable effect on oocyte swelling [61]. Furthermore, the crystal structure of hAQP4 shows the channel in an open formation in the absence of phosphorylation at S111 [74], suggesting that phosphorylation of S111 alone does not explain an increase in water permeability found in the presence of lead, glutamate, and potassium [67,68,69]. The contrast between results found in mammalian cells, which suggest phosphorylation of either S111 or S180 mediates an increase or decrease, respectively, in water permeability without internalization of the protein [67,68,69,70], and results found in oocytes, which show no effect of S111 phosphorylation and protein internalization when S180 undergoes phosphorylation [61,64], suggests that the pathways activated by phosphorylation of either S111 or S180 may be different in oocytes and mammalian cells.

Phosphorylation of AQP4 remains a complex topic. While most current knowledge focuses on S111, S180, and S276, there remain many more proposed phosphorylation sites, including S285, S315, S316, S321, and S322 (Figure 1), the functions of which have yet to be determined [56,62], though experiments with mutant rAQP4 proteins in oocytes suggest that phosphorylation of the COOH-terminal serine residues S315, S316, S321, and S322 does not have any effect on either trafficking or channel gating [75]. It is worth noting that the S315 phosphorylation site identified in rats is not conserved in human AQP4 [62], as the serine is instead a glutamine (Figure 1, white residue). The effect of phosphorylation on AQP4 remains an intriguing field of study, with many mechanisms and pathways as of yet undiscovered.

## 4. Metal Ion Mediated Regulation of AQP4

The majority of aquaporins are modulated by at least one metal ion [2,76,77]. Mercury especially has been shown to have an inhibitory effect on almost all AQPs, with the exception of AQP6 to which it has an activating effect [78]. Interestingly, when AQP4 was first characterized in oocytes, the name originally given to it was “mercurial-insensitive water channel”, as rAQP4 expressed in oocytes uniquely showed no change in activity when exposed to Hg^2+^ [4]. Experiments in proteoliposomes however, revealed the truth, the binding site for Hg^2+^ on AQP4 is intracellular, meaning that in the case of oocytes and most cell models, Hg^2+^ cannot reach its direct binding site on AQP4 and can therefore not modulate its activity (Figure 1, dark blue residues) [79,80]. In the proteoliposome model, approximately 50% of the AQP4 will have been inserted into the membrane in a reverse position to in vivo, exposing the Hg^2+^ binding site to the external solution, allowing for inhibition to occur [79]. This binding site was identified via site specific mutagenesis to the C178 residue in loop D of the intracellular domain of rAQP4, and the effect was reversible after treatment with β-mercaptoethanol, suggesting that Hg^2+^ binds covalently to this residue, and may cause conformational changes in AQP4 [76,79]. A site specific mutation at the C253 residue of AQP4 also reduced the inhibitory capacity of Hg^2+^, but not to the same extent as the C178 mutation [79]. In vivo testing of acute methylmercury (MeHg) exposure in marmosets caused a significant increase in expression of marmoset AQP4 mRNA in the frontal lobe, occipital lobe, and cerebellum, and of AQP4 protein in the occipital lobe and cerebellum, the majority of which was located in reactive astrocytes as determined with a glial fibrillary acidic protein (GFAP) AQP4 double stain [81]. It appears that even as mercury is able to inhibit AQP4 if it can access C178, prolonged mercury exposure in vivo actually leads to an upregulation of AQP4.

A second metal ion which appears to bind to the same C178 residue is zinc [82]. Experiments on rAQP4 M23 proteoliposomes showed that ZnCl_2_ at a concentration of 1000 μM was able to inhibit water permeability of AQP4 by approximately 30%, and the addition of a thiol oxidizing agent diamide (1 mM) enhanced the inhibitory effect, decreasing the concentration needed for 30% inhibition to 20 μM [82]. In addition to diamide, the general anesthetic propofol was shown to drastically increase the inhibitory effect of zinc on hAQP4 via interaction with both zinc and C253 in the C-terminal tail of AQP4 [83]. This inhibitory effect was specific to hAQP4, as hAQP1 proteoliposomes showed no reaction post incubation with zinc and propofol [83]. In the same paper which first identified zinc as an inhibitor of rAQP4 in proteoliposomes, copper was also identified to inhibit rAQP4 [82]. Prior to those experiments, copper inhibition was thought to be specific to AQP3, as tests in human bronchial epithelial cells (BEAS-2b) which were genetically modified to express several AQPs did not find any significant difference in cells exposed to CuCl_2_ when expressing AQP4 and AQP7 [84]. This is likely due to Cu^2+^ inhibiting AQP4 via the same mechanism as mercury and zinc, by binding to the C178 residue of AQP4, which is not accessible in oocyte or cell models where Cu^2+^ is unable to cross the cell membrane [80,82].

Not all metal ions which regulate AQP4 activity act via direct binding, some act via indirect means, triggering a phosphorylation pathway which leads to changes in AQP4 expression or water permeability. As briefly mentioned above, lead exposure increased water permeability of cultured rat astrocytes by 40% via activating a CaMKII pathway leading to the phosphorylation of S111 [69], although in vivo testing in rats treated with lead acetate did not show an increase in rAQP4 mRNA even when elevated levels of lead could be detected [69]. Cultured rat astrocytes treated with 25 μM manganese also showed an increase in water permeability, with a corresponding time-dependent increase in rAQP4 expression in the plasma membrane [85]. This increase in rAQP4 expression in the plasma membrane was not accompanied by an increase in rAQP4 mRNA, implying that increase in expression was due to either higher stability of the protein or migration to the membrane, and not due to an increase in protein synthesis [85]. The effect of manganese on astrocytes was attributed to an activation of the mitogen-activated protein kinases (MAPK) extracellular signal-regulated kinase (ERK)1/2 and p38-MAPK, as these kinases were found to have an increased level of phosphorylation and inhibitors of these kinases blocked the manganese-induced increase in rAQP4 protein levels [85]. A second metal ion which activates the MAPK pathway, leading to increased mAQP4 expression is ferrous iron (Fe^2+^) [86]. Treatment with 25 μM Fe^2+^ significantly increased mAQP4 expression, and this change was eliminated by MAPK inhibitors and reduced by treatment with antioxidants [86]. Interestingly, mAQP4 gene silencing via transfection of mAQP4 siRNA (100 nM) not only reduced mAQP4 expression, but also protected astrocytes from Fe^2+^-induced cell death, as astrocytes treated with Fe^2+^ and control siRNA showed significant levels of cell death in comparison, which suggests that AQP4 plays a key role in Fe^2+^-induced apoptosis [86]. Experiments with iron in vivo are mostly related to the study of AQP4 expression in intracerebral hemorrhage (ICH), as iron overload commonly follows ICH due to red blood cell lysis releasing large amounts of iron in the brain [87,88]. In these experiments, iron accumulation was detected 1 day post ICH, and peaked at 7 days post ICH. This increase in iron is accompanied by an increase in AQP4 expression, peaking at 3 days and maintained until day 7 [89]. Treatment with an iron chelator, deferoxamine (DFO), significantly reduced the iron overload, brain water content, and rAQP4 levels, indicating that the increase in rAQP4 and changes in brain water level are indeed a reaction to the increased levels of Fe^2+^ [89]. In a mouse model of ICH, treatment with curcumin (turmeric) reduced the cerebral edema post ICH via decreasing the gene and protein expression of mAQP4 and mAQP9 [90]. Via testing in vitro in cultured mouse astrocytes, this study demonstrated that rather than the MAPK pathway being activated by Fe^2+^, it is instead suggested that the increase of mAQP4 and mAQP9 expression is a result of an increased expression and nuclear translocation of Nuclear Factor κB (NF-κB) p65, a regulator of transcription of several genes involved in immune and inflammatory responses [91], as the increased expression was dose-dependent to Fe^2+^ and treatment with either curcumin or NF-κB inhibitors interfered with the NF-κB pathway and inhibited the increase in protein concentration of mAQP4 and mAQP9 [90]. It is possible that both the MAPK and NF-κB pathways contribute to the increase in AQP4 expression in response to Fe^2+^ overload.

In general, it appears that metal ions which are able to bind directly to AQP4 often cause an inhibitory effect, decreasing the water permeability of the water channel itself [76,79,80,82,83], whereas an increase in the protein expression levels of AQP4 appears to be a common reaction in astrocytes to high levels of metal ions in their environment [69,76,81,85,86,89]. An overview of the metal ions discussed in this section and the small molecule inhibitors discussed in Section 5 can be found in Table 2.

## 5. Regulation of AQP4 via Small Molecule Inhibitors

A small molecule inhibitor is a compound which may act as an inhibitor of AQP4 without the inclusion of a metal ion in the compound. The identification of small molecule AQP4 antagonists could be extremely valuable, as AQP4 activity or expression is implicated in several pathological conditions [12,20,21,32,36,92,93], and a small molecule inhibitor would allow for inhibition of AQP4 without the need to factor in metal ion toxicity. In 2007, several arylsulfonamides were identified in silico as potentially having an inhibitory effect on rAQP4 M23, three of which showed said effect in oocytes expressing hAQP4 M23 with fairly low concentrations (1–20 μM) required; Acetazolamide (AZA, ~80% inhibition), 6-Ethoxybenzothiazole-2-sulfonamide (EZA, ~68% inhibition), and 4-acetamidobenzulfonamide (~23% inhibition) [94]. These results were rapidly followed up in the same year with a screening of several other compounds in the same manner, including clinically approved anti-epileptic drugs and several other compounds identified in silico to potentially bind to rAQP4 M23 [95,96]. Of the total 32 compounds tested between these three studies, 24 showed a significant inhibitory effect when tested at 20 μM in vitro in oocytes expressing hAQP4 M23, with significant inhibition ranging from 9%–80%, with acetazolamide identified as the most potent inhibitor [94,95,96]. There is some controversy as to whether or not the compounds identified in the three aforementioned studies actually inhibit AQP4. Two of these compounds, acetazolamide and valproic acid, which had shown inhibition rates of 80% and 40% in the oocyte experiments, were tested in proteoliposomes containing rAQP4 M23, and only acetazolamide was found to reversibly inhibit rAQP4 M23 at a concentration of 1.25 mM, whereas valporic acid at a concentration of 3 mM showed no inhibitory effect on the proteoliposomes [97]. Interestingly the half maximal inhibitory concentration (IC_50_) of AZA in proteoliposomes was approximately 1.1 mM and max inhibition was 53.3%, which is in stark contrast to the IC_50_ reported in oocytes of 0.86 μM and max inhibition of 80% [94,97]. The difference in results between proteoliposomes and oocytes may be due to the use of rAQP4 M23 in the proteoliposome testing, and hAQP4 M23 in the oocyte screening, in addition to the difference in incubation times between studies (30 min for proteoliposomes, 2–3 h for oocytes) [94,96,97]. A study published in 2008 tested eight of the most potent proposed AQP4 inhibitors (AZA, Acetylsulfanilamide (ASA), EZA, Topiramate, Zonisamide, Phenytoin, Lamotrigine, and Sumatriptan, which had inhibition levels of 80%, 66%, 67%, 48%, 58%, 54%, and 54% in oocyte assays respectively [94,95,96]) and found no inhibitory activity of any of the compounds [98]. The eight compounds were tested using four different methods with three different cell types for determining water permeability; A stopped-flow assay using AQP4-expressing Fischer rat thyroid (FRT) cell plasma membrane vesicles, a transepithelial water transport assay using the same AQP4-expressing FRT cells, a glial cell water permeability assay using glial cells from mAQP4^+/+^ and mAQP4^−/−^ mice, and a stopped flow assay using mouse erythrocytes [98]. These results led to the conclusion that the methods used in the original three studies were flawed, and none of the identified compounds can be considered inhibitors of AQP4 [98]. It is worth noting, that in both studies which refuted the findings in the original three papers, human AQP4 M23 was not used, and either rat or mouse AQP4 M23 was used instead [97,98]. An in silico study of acetazolamide bound to the 3D structure of rAQP4 M23, identified ten residues (T56, G146, V147, T148, T149, H151, I205, G209, A210, and R216) as the most likely candidates for binding of acetazolamide to rAQP4 M23, and subsequent inhibition of water permeability (Figure 1, green residues) [99]. Of these ten residues identified in silico to bind to rAQP4 M23, all but one are conserved in hAQP4, as T149 in rAQP4 is actually M149 in hAQP4 (Figure 1, white residues), which may partially explain the variability in results when using hAQP4 and rAQP4.

While in vitro assessments of some of these compounds have shown conflicting results [94,95,96,97,98], a few of the compounds have been tested in vivo in models of various neurological conditions. AZA has been shown to protect against edema formation in mouse models of traumatic brain injury (TBI), preventing the loss of mAQP4 polarization and reducing cytotoxic edema in cell bodies within the cortex and hippocampus of AZA treated mice [100]. These findings are backed up by tests in an in vitro astrocyte model of TBI, showing reduced mAQP4 expression, reduced S100B expression, and reduced cell death in cells exposed to AZA when compared to untreated cells [101]. AZA has also been shown to protect against dysfunction of articular chondrocytes in adjuvant-induced arthritis (AIA) rats by protecting against the overexpression of rAQP4 which corresponds to an increase in symptom severity and joint damage pathological scores in this model [102]. AZA treatment in this study decreased rAQP4 protein level, increased cell proliferation, and increased mRNA levels of type-II collagen (COII) and aggrecan, suggesting that AZA is able to partially normalize the dysfunction of AIA articular chondrocytes [102]. Two non-specific inhibitors of AQP4; AZA and tetraethylammonium (TEA), were tested in hippocampal mouse brain slices and induced an increase in S100B secretion [103], a protein secreted by astrocytes and implicated in astroglial activation and cell injury [103,104].

Another compound identified as a potent hAQP4 M23 inhibitor in oocytes (73% max inhibition, IC_50_ of 3 μM) that has been evaluated in vivo is 2-(Nicotinamide)-1,3,4-thiadiazole, or TGN-020 [17,96]. In silico studies using an rAQP4 M23 model identified the likely binding sites of TGN-020 as D69, M70, I73, F77, V145, I205, and R216 (Figure 1, purple residues), one of which is not conserved in hAQP4 (V145 is L145 in hAQP4. Figure 1, white residues) [96]. TGN-020 was evaluated in a mouse model of focal cerebral ischemia, and pretreatment of TGN-020 significantly reduced the size of ischemia related brain edema, accompanied with a significant decrease in brain swelling in the TGN-020 treatment group [17]. TGN-020 was also shown to increase regional cerebral blood flow via inhibition of AQP4 in mice [105]. Both mAQP4^+/+^ and mAQP4^−/−^ mice were treated with TGN-020, and live imaging was used to determine the relative cerebral blood flow. TGN-020 induced an increased blood flow in mAQP4^+/+^ mice, but no increased flow in mAQP4^−/−^ mice, implying that the TGN-020 is acting on mAQP4 [105]. Furthermore, AZA treatment in the same mice caused a significant increase in blood flow in both wild type and knockout mice, confirming that AZA may not be mAQP4 specific, whereas a saline injection did not increase blood flow in either type of mouse [105]. A single dose of TGN-020 (intraperitoneal 100 mg/kg) administered 15 min post medial cerebral artery occlusion in a rat model of non-reperfusion ischemia significantly reduced edema size, glial scar, albumin effusion, and apoptosis, which were attributed to inhibition of rAQP4, although no rAQP4 knockout rat model was tested to confirm [106].

While the in vitro screening of small molecule inhibitors remains a controversial field, the increasing number of publications which indicate an effect of some of these molecules in vivo shows that there is a possibility that some of these compounds may actually inhibit AQP4. With that being said, it is entirely possible that treatment of in vivo models with these small molecule inhibitors may cause inhibition or loss of AQP4 expression due to indirect methods, rather than via direct binding, as in vitro methods tend to attempt to detect. Regardless, the potential of a non-toxic AQP4 specific inhibitor in several neurological conditions makes this an intriguing field not only in theoretical research but also in the medical field.

## 6. Conclusions

Regulation of AQP4 in the central nervous system remains a complicated and intricate topic, with many potential interactions and a large amount of model based variability. AQP4 appears to be regulated via many pathways, from gene silencing/activation to protein modification, and via direct binding of ions or compounds. Recently more and more studies are revealing different aspects of a larger picture in AQP4 regulation, although it is necessary to continue to develop this field of research, as many questions remain unanswered. A clear understanding of AQP4 regulation is required, as AQP4 has an important role both in the normal function and dysfunction of the CNS. The role of AQP4 in several CNS disorders has heightened the importance of identifying a non-toxic AQP4 activity modulator which could be used for therapeutic applications. The main challenges facing development of AQP4 targeting therapeutic agents are selectivity, toxicity, functional inhibition, and delivery into the CNS. Metal ions, while effective, are often too toxic to be used. The conflicting results shown when using small molecule inhibitors showcase the difficulty in identifying an effective activity modulator. Both a phosphorylation based approach and a microRNA based approach show promise, but with such complex pathways, disruption of the status quo could lead to unforeseen consequences. While several antibody based therapies have been developed for application in NMO, only one specifically targets AQP4, Aquaporimab, and this drug has not been tested in humans, nor is it designed to have an effect on water permeability through AQP4. At this point in time, the most promising AQP4 specific activity modulator appears to be TGN-020, as it has demonstrated selectivity, functional inhibition, and successful delivery to the CNS in vivo, though only during studies with one single dose just prior to analysis, so whether or not sustained therapeutic concentration is possible has not been established. Even though a vast amount of information has recently been revealed regarding AQP4, there remains a need to continue to study the regulatory mechanisms of this molecule, and a need to standardize the analysis methods used in order to combat the inter-method variability present in this field.

## Figures and Tables

**Figure 1 ijms-21-01603-f001:**
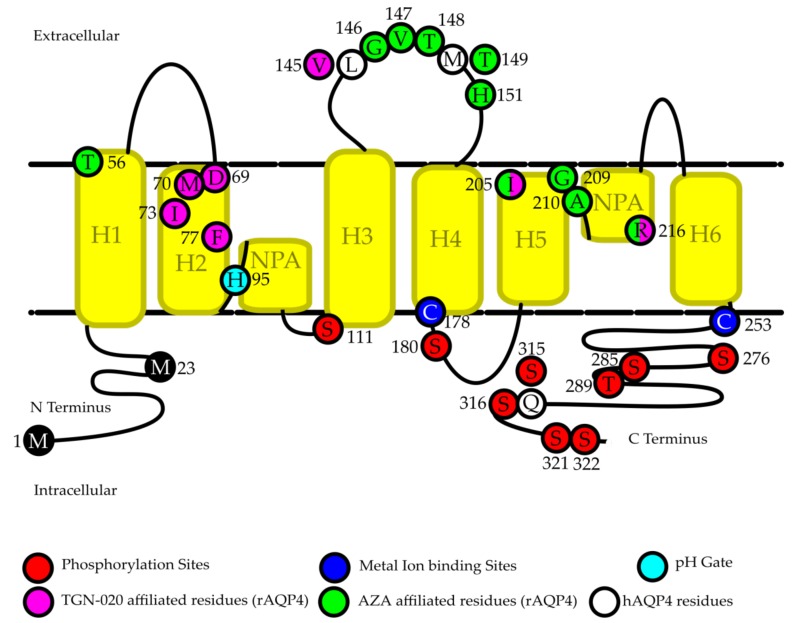
Important residues in AQP4 regulation. The location and function of several important residues identified to have a role in AQP4 regulation. White residues indicate the human equivalent to sites which were identified in rat AQP4 (rAQP4) as having important regulatory functions but are not conserved in human AQP4 (hAQP4). Numbering is based off of human AQP4.

**Table 1 ijms-21-01603-t001:** A summary of known MicroRNAs which have an effect on Aquaporin-4 (AQP4) gene expression, including which species of AQP4 the microRNAs (miRNAs) is known to be affiliated with, the proteins targeted by the miRNA, and the role each miRNA has in endogenous conditions.

MicroRNA	AQP4 Species	Effect	Role	Reference
miR-224/miR-19a	Mouse, Rat	Downregulates AQP4 and Cx43	Astrocyte connectivity and water permeability.	[48]
miR-29b	Mouse	Downregulates AQP4	Reaction to ischemia, reduces infarct volume edema and BBB disruption. Upregulated in response to 1,2-DCE exposure, leading to an induction of brain edema.	[49,50]
miRNA-145	Rat	Downregulates AQP4	Reaction to ischemia, Attenuates AQP4 induced astrocyte injury.	[51]
miRNA-320a	Mouse, Human, Rat	Downregulates AQP4 and AQP1	Increases infarct volume in ischemic cerebral edema, inhibits glioma cell invasion and migration.	[52,53]
miRNA-130a	Human, Rat	Downregulates AQP4 M1	Increases infarct volume in ischemic cerebral edema.	[54]
miRNA-130b	Rat	Downregulates AQP4	Reaction to ischemia, Attenuates AQP4 induced astrocyte injury.	[55]

**Table 2 ijms-21-01603-t002:** An overview of the metal ions and small molecule inhibitors discussed in this review, including the species of AQP4 tested, the effect the molecule had in in vitro testing, the effect the molecule had in in vivo testing (where applicable), and the proposed mechanism of action of the molecule.

Compound	Species	In Vitro Effect	In Vivo Effect	Mechanism	Reference
Mercury(Hg^2+^, MeHg)	Rat, Marmoset	Inhibition of water permeability through AQP4 in proteoliposomes. No effect in oocyte model.	Upregulation of AQP4 expression in reactive astrocytes.	Inhibition mediated via binding at C178 and C253	[76,79,81]
Zinc (ZnCl_2_)	Rat, Human	Inhibition of water permeability through AQP4 in proteoliposomes, increased in presence of propofol and diamide.	Not applicable.	Inhibition mediated via binding at C178 and C253	[82,83]
Copper (CuCl_2_)	Rat	Inhibition of water permeability through AQP4 in proteoliposomes. No effect in BEAS-2b cell model.	Not applicable.	Inhibition mediated via binding at C178	[82]
Lead (Pb^2+^)	Rat	Increase in water permeability through AQP4.	No increase in AQP4 expression.	CaMKII induced phosphorylation of S111	[69]
Manganese (Mn^2+^)	Rat	Increase in AQP4 expression on plasma membrane without increase in overall protein expression.	Not applicable.	Activation of ERK1/2 and p38-MAPK.	[85]
Ferrous Iron (Fe^2+^)	Mouse, Rat	Increase in AQP4 protein expression.	Increase in AQP4 expression in ICH models.	Both activation of the MAPK pathway and the NF-κB pathway likely contribute to the increased AQP4.	[86,87,88,89,90,91]
Acetazolamide	Mouse, Rat, Human	Potent inhibition of hAQP4 in oocytes. Weak inhibition of rAQP4 in proteoliposomes. No inhibition in FRT cell vesicles, mouse glial cells, and mouse erythrocytes.	Protects against edema during TBI. Alleviates AIA symptoms via suppression of AQP4 expression.	Direct binding inhibition likely mediated by binding to T56, G146, V147, T148, T149 (rat)/M149 (human), H151, I205, G209, A210, and R216. Indirect mechanism unidentified.	[94,95,96,97,98,99,100,101]
Valporic acid	Rat, Human	Potent inhibition of hAQP4 in oocytes. No inhibition of rAQP4 in proteoliposomes.	Not applicable.	Mechanism unidentified.	[96,97]
EZA, Topiramate, Zonisamide, Phenytoin, Lamotrigine, Sumatriptan	Mouse, Rat, Human	Potent inhibition of hAQP4 in oocytes. No inhibition in FRT cell vesicles, mouse glial cells, and mouse erythrocytes.	Not applicable.	Mechanism unidentified.	[94,95,96,98]
TGN-020	Mouse, Rat, Human	Potent inhibition of hAQP4 in oocytes.	AQP4 specific inhibition leading to improved outcomes of ischemic stroke and increased regional blood flow.	Direct binding inhibition likely mediated by binding to D69, M70, I73, F77, V145 (rat)/L145 (human), I205, and R216.	[17,96,105,106]

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
