# Peer review of "Regulation of AQP4 in the Central Nervous System"

_ijms, 2020, doi:10.3390/ijms21051603_

Round 1

Reviewer 1 Report

This is a very well written review on Acquaporin 4 and its functions in the CNS, describing accurately all the possible functional changes induced on the molecule by a variety of chemical changes, phosphorylations, ions, miRnas, small molecule inhibitors etc..

The review focuses very well on the different reflections on functionality induced by these agents on Acquaoporin 4 and underlines as the clinical effects may be completely different according to the type of chemical interaction of the various molecules and substances on its functon.

Due to the important possible drawbacks of this astonishing amount of data on this molecule, the paper is to be considered timely, appropriate and worth publishing.

Author Response

Response to reviewer 1 comments:

This is a very well written review on Acquaporin 4 and its functions in the CNS, describing accurately all the possible functional changes induced on the molecule by a variety of chemical changes, phosphorylations, ions, miRnas, small molecule inhibitors etc..

The review focuses very well on the different reflections on functionality induced by these agents on Acquaoporin 4 and underlines as the clinical effects may be completely different according to the type of chemical interaction of the various molecules and substances on its functon.

Due to the important possible drawbacks of this astonishing amount of data on this molecule, the paper is to be considered timely, appropriate and worth publishing.

Response 1: Thank you for taking the time to review our article and thank you for your kind words.

Reviewer 2 Report

The manuscript reviews the role of AQP4 in the central nervous system and implications in several CNS disorders, describing the different types of AQP4 regulation that may have an impact on disease or used to develop new therapeutics. The manuscript is well written, presents interesting information and compiles recent studies on the topic. Some comments to improve this review:

- Given the diverse number of inhibitors and effects reported (metals and small molecules), the authors should prepare and include a new table assembling all the information (inhibitor, effect, mechanism if known), similar to the already presented for miRNAs. This would help the reading and allow summarizing all the information.

- Elaborate conclusions in a more descriptive manner. Are there any AQP4 modulators in clinical trials? Specifically, which is their potential to be developed as new therapeutics?

Author Response

Response to Reviewer 2 Comments:

The manuscript reviews the role of AQP4 in the central nervous system and implications in several CNS disorders, describing the different types of AQP4 regulation that may have an impact on disease or used to develop new therapeutics. The manuscript is well written, presents interesting information and compiles recent studies on the topic. Some comments to improve this review:

Point 1: Given the diverse number of inhibitors and effects reported (metals and small molecules), the authors should prepare and include a new table assembling all the information (inhibitor, effect, mechanism if known), similar to the already presented for miRNAs. This would help the reading and allow summarizing all the information.

Response 1: We appreciate the suggestion.  A new table has been prepared and inserted at the end of Section 5 of the article.  This table gives an overview of all ions and compounds discussed in Sections 4 and 5 of the article, including the inhibitor being discussed, the species of AQP4 which has been tested with this inhibitor, the effect the inhibitor had in in vitro studies, the effect the inhibitor had in in vivo studies (where applicable), the proposed mechanism of action (if known), and a reference to the studies in which this inhibitor is discussed.

Point 2: Elaborate conclusions in a more descriptive manner. Are there any AQP4 modulators in clinical trials? Specifically, which is their potential to be developed as new therapeutics?

Response 2: The conclusions have been reorganized and expanded upon with the authors' opinion on the potential of AQP4 modulators to be developed as new therapeutics, focusing on the 4 main regulatory categories discussed in the review.  

Round 2

Reviewer 2 Report

The authors replied to all comments and improved the manuscript.